# Infectivity of SARS-CoV-2 on Inanimate Surfaces: Don’t Trust Ct Value

**DOI:** 10.3390/ijerph192417074

**Published:** 2022-12-19

**Authors:** Johannes K. Knobloch, Susanne Pfefferle, Marc Lütgehetmann, Dominik Nörz, Eva M. Klupp, Cristina E. Belmar Campos, Stefan Kluge, Martin Aepfelbacher, Birte Knobling, Gefion Franke

**Affiliations:** 1Institute for Medical Microbiology, Virology and Hygiene, Department for Infection Prevention and Control, University Medical Center Hamburg-Eppendorf, Martinistr. 52, 20246 Hamburg, Germany; 2Institute for Medical Microbiology, Virology and Hygiene, University Medical Center Hamburg-Eppendorf, Martinistr. 52, 20246 Hamburg, Germany; 3Department of Intensive Care Medicine, University Medical Center Hamburg-Eppendorf, Martinistr. 52, 20246 Hamburg, Germany

**Keywords:** SARS-CoV-2, surface contamination, quantitative PCR, Ct value, disinfection

## Abstract

SARS-CoV-2 RNA is frequently identified in patient rooms and it was speculated that the viral load quantified by PCR might correlate with infectivity of surfaces. To evaluate Ct values for the prediction of infectivity, we investigated contaminated surfaces and Ct-value changes after disinfection. Viral RNA was detected on 37 of 143 investigated surfaces of an ICU. However, virus isolation failed for surfaces with a high viral RNA load. Also, SARS-CoV-2 could not be cultivated from surfaces artificially contaminated with patient specimens. In order to evaluate the significance of Ct values more precisely, we used surrogate enveloped bacteriophage Φ6. A strong reduction in Φ6 was achieved by three different disinfection methods. Despite a strong reduction in viability almost no change in the Ct values was observed for UV-C and alcoholic surface disinfectant. Disinfection using ozone resulted in a lack of Φ6 recovery as well as a detectable shift in Ct values indicating strong degradation of the viral RNA. The observed lack of significant effects on the detectable viral RNA after effective disinfection suggest that quantitative PCR is not suitable for predicting the infectivity of SARS-CoV-2 on inanimate surfaces. Ct values should therefore not be considered as markers for infectivity in this context.

## 1. Introduction

The pandemic following the emergence of severe acute respiratory syndrome coronavirus 2 (SARS-CoV-2) causing coronavirus disease 2019 (COVID-19) continues to be sustained today by different virus variants [1,2]. As with other viral pathogens of the upper respiratory tract, the main routes of transmission of SARS-CoV-2 are droplet infection during person-to-person contact and aerosol-borne infection originating from index persons with a particularly high viral load [3,4]. In addition, indirect transmission via inanimate surfaces is discussed for all viruses that cause respiratory infections [4,5]. Touching surfaces contaminated with SARS-CoV-2 may therefore be a potential source of viral transmission. However, the proportion of this route of transmission is difficult to estimate.

Several studies have demonstrated that SARS-CoV-2 can remain infective on surfaces up to several days in experimental settings [6]. Depending on the surface properties, SARS viruses enriched in cell cultures can be recovered from surfaces at very high initial concentrations for hours to days. Additionally, other studies have shown that SARS-CoV-2 RNA can be detected in the patient environment [7]. However, these studies employed different sampling methods as well as different extraction procedures and the limit of detection (LOD) was not reported in all studies. Despite these limitations of the studies investigating environmental contamination by PCR it was hypothesized that a Ct value from a surface can be used to estimate infectivity [8]. However, in nine studies reporting the attempt to recover SARS-CoV-2 virus from PCR-positive hospital surfaces in cell culture only two studies reported successful virus isolation [7].

Therefore, this study aimed to increase the understanding of the importance of Ct values in assessing the infectivity of viruses on inanimate surfaces. The surfaces of patient rooms in an intensive care unit (ICU) for COVID-19 patients as well as artificially SARS-CoV-2-contaminated surfaces were evaluated by PCR and cell culture. In addition, the change in Ct values for SARS-CoV-2 and the surrogate virus bacteriophage Φ6 was studied after disinfection effective against enveloped viruses.

## 2. Materials and Methods

### 2.1. Environmental Sampling

Samples from the immediate patient environment in an ICU were collected with moistened flocked swabs (eSwab^®^ 480CE + screw-cap tube filled with 1 mL of Liquid Amies Medium, Copan, Brecia, Italy). In non-ventilated awake patients, three surfaces frequently touched by the patients themselves (control panel of the bed, telephone and nightstand) as well as the space bar and the mouse of the patient-related computers were swabbed. In ventilated sedated patients, three surfaces frequently touched by medical staff were sampled in addition to the space bar and mouse of the patient-related computer. After thorough vortexing of the swab in the Amies Medium, 200 µL of the medium was taken to be further examined in viral culture if the PCR result was positive.

### 2.2. Quantitative RT-PCR

Detection and quantification of SARS-CoV-2 RNA was performed fully automated on the Cobas 6800 platform using either a lab-developed test (LDT) [9] or the commercially available Roche SARS-CoV-2 IVD assay according to the manufacturer´s instructions. For the SARS-CoV-2 IVD assay, a target value of 2 Ct was used for quantification as described in [10]. For the real-time PCR with detection of Φ6, primer sequences were used as originally described by Gendron et al. [11] targeting the Φ6 S1 gene (coding for the P8 protein) located on the S segment.

### 2.3. Cell Culture and Virus Isolation

Vero cells (ATCC^®^ CRL-1586) were cultivated in DMEM supplemented with 10% FCS, 1% Penicillin/ Streptomycin, 1% L-Glutamine, (200 mM), 1% Sodium pyruvate and 1% non-essential amino acids (all Gibco/Thermo Fisher, Waltham, MA, USA) under standard conditions. For virus isolation attempts, cells were seeded into 24-well plates (TPP, Trasadingen, Switzerland) at 80–90% confluence. A 250 µL volume of the samples was added to each well and incubated at 37 °C for 1 h for virus adsorption. Thereafter, 1 mL of fresh cell-culture medium was added. Cells were monitored daily for cytopathic effect. Absence of virus growth was confirmed by quantitative RT-PCR.

### 2.4. Contamination of Surfaces with Patient Specimen and Virus Recovery

The liquid portion of bronchioalveolar lavage (BAL, anonymized at the source) with confirmed SARS-CoV-2 RNA detection was used to contaminate inanimate surfaces. In order to be able to soil a sufficient number of surfaces, the materials were diluted 1:14 maximum with 0.85% NaCl with 0.03% bovine serum albumin. Of the diluted BAL specimens, 50 µL was streaked on ceramic tiles (5 × 5 cm, #3709PN00, Villeroy&Boch, Mettlach, Germany) using a single-use inoculation spreader (Sarstedt, Nürnbrecht, Germany) and dried. To recover the material, the surface of a total volume of 2 mL was rinsed 15 times with 1 mL universal transport medium (UTM; Miraclean Technology, Shenzhen, China) after the individual disinfection process or at the control time points without disinfection.

### 2.5. Contamination of Surfaces with Bacteriophage Φ6 and Quantification after Recovery

For the comparison of infectivity and Ct value, ceramic tiles were contaminated with the enveloped bacteriophage Φ6 as a surrogate virus of SARS-CoV-2. Phage Φ6 (DSM 21518) and the bacterial host strain *P. syringae* pv. *syringae* (DSM 21482) were purchased from Leibniz-Institute DSMZ—Deutsche Sammlung von Mikroorganismen und Zellkulturen GmbH (Braunschweig, Germany). Initial lysate of Φ6 with a titer of 4 × 10^11^ plaque-forming units (pfu)/mL was produced using a top agar overlay technique. Then, 20 µL of a 1:10 dilution was deposited onto ceramic tiles resulting in an initial concentration of >107 pfu/mL on each carrier. After each decontamination process, Φ6 from both treated and untreated carriers was recovered by rinsing off the surface with 1ml Tryptic Soy Broth (TSB) + 5 mM CaCl_2_ medium 15 times and transferred to a reaction tube. For direct quantification of viral concentration, a plaque assay was performed. Tenfold serial dilutions of the rinsing fluid (100 µL) were mixed with a fresh culture of the host strain as well as 4 mL soft agar with subsequent transfer on Tryptic Soy Agar (TSA) + 5 mM CaCl_2_ culture media. Plates were incubated at 23 °C for 24 h. Finally, plaque-forming units were determined and reduction factors were decided.

### 2.6. Disinfection of Artificially Contaminated Surfaces with Ozone

An automatic room-disinfection system (STERISAFE Pro version 1.0, STERISAFE ApS, Copenhagen, Denmark) generating ozone was used to disinfect surfaces contaminated with patient material or phage Φ6 in independent test runs. In accordance with the manufacturer’s instructions, a decontamination time of 60 min with an ozone concentration of 80 ppm and an average relative humidity of 90% was used (total time of up to 2 h). The decontamination process was performed in a 6 m^3^ gas-tight test room with carriers placed horizontally on a shelf board in three independent experiments. In order to model the disinfection of easily accessible sites, two contaminated surfaces per pathogen were disinfected without any protection. To model hard-to-reach sites, two carriers each were disinfected in a gas-tight 0.35-L box (Emsa, Emsdetten, Germany) with two 3-mm holes. Furthermore, in each experiment two contaminated control carriers were placed in a room without treatment and processed at the end of the experiment (T_eoe_) together with the disinfected carriers. As the disinfection procedure using ozone took the longest time period two additional carriers were investigated before starting the experiments to quantify the burden of contamination at time point zero (T0) in order to exclude spontaneous reduction over time.

### 2.7. Disinfection of Artificially Contaminated Surfaces with UV-C

Surfaces contaminated with patient material or phage Φ6 were placed horizontally at a distance of 0.5 m and 1.0 m from a UV-C light source (UVD-robot Model B, UVD Robots ApS, Odense, Denmark) resulting in UV intensities of 50 mJ/cm^2^ and 200 mJ/cm^2^, respectively. This radiation exceeded the doses reported to have virucidal effectivity on dry surfaces and in direct radiated fluids (Appendix A). UV intensity was measured with UV indicators (UCV Dosimeter, UVD Robots ApS, Odense, Denmark) placed directly beside the contaminated surfaces. Two untreated contaminated controls were carried out along with each experiment. Experiments were conducted three times with duplicate surfaces.

### 2.8. Disinfection of Artificially Contaminated Surfaces with an Alcohol-Based Disinfectant

Surfaces contaminated with patient material or phage Φ6 were evenly covered with 20 µL of a commercial alcoholic surface disinfectant tested as virucidal against enveloped virus (antifect N liquid, Schülke, Norderstedt, Germany; containing 25% wt/wt ethanol [94%] and 35% wt/wt propan-1-ol). The disinfectant was spread with a single-use inoculation spreader (Sarstedt, Nürnbrecht, Germany) and dried completely. The drying time was at least 30 s reaching full virucidal activity against enveloped viruses, according to the manufacturer’s instructions.

## 3. Results

For 20 patients with quantified detection of SARS-CoV-2 RNA in the upper airways, viral RNA was detected on 37 of 143 investigated surfaces (Figure 1). In patients with high RNA load in their airways, surfaces were contaminated more frequently. Most surfaces detected to be PCR-positive showed a low load of viral RNA. Virus isolation was attempted for six surfaces that tested positive for SARS-CoV-2 RNA with the relatively highest measurable RNA load (mean 47,258, range 2543–158,241 copies/mL). However, virus isolation failed for these surfaces.

In a second step, we artificially contaminated surfaces with a liquid portion of bronchioalveolar lavage (BAL) from four different patients with confirmed SARS-CoV-2 RNA detection and performed PCR directly or after disinfection with ozone, UV-C or a commercial alcoholic surface disinfectant. Disinfection with ozone showed a stronger Ct-value shift on both openly accessible and hard-to-reach surfaces, compared to effective doses of UV-C or alcoholic disinfection (Figure 2A, Appendix A).

Since differences between Ct values (Δ_Ct_) can be extrapolated into the number of copies of viral RNA, a reduction factor could be derived (Appendix A). Storage of the surfaces for more than two hours showed no reduction in SARS-CoV-2 viral RNA (Δ_Ct_ 0.02 to 0.1; log reduction 0 to 0.033). Disinfection with ozone showed a higher reduction in viral RNA regardless of whether the surfaces were exposed directly (Δ_Ct_ 6.20 to 7.50; log reduction 1.518 to 1.835) or difficult to access in a closed container with only two small holes (Δ_Ct_ 3.95 to 6.13; log reduction 0.955 to 1.572) compared to intensive UV-C radiation with 200 mJ/cm^2^ (Δ_Ct_ 2.43 to 3.12; log reduction 0.833 to 1.073). Surface disinfection with a commercial alcoholic disinfectant without mechanical cleaning displayed the lowest calculated reduction (Δ_Ct_ 0.62 to 2.25; log reduction 0.181 to 0.704). However, as observed for surfaces in the patient rooms, SARS-CoV-2 could not be cultivated from swabs taken from surfaces contaminated with the fluid from BAL.

In order to determine the significance of PCR results for the assessment of infectivity more precisely, the easily culturable surrogate, enveloped bacteriophage Φ6, was investigated. It could be shown that except for the UV-C dose of 50 mJ/cm^2^, a strong reduction in Φ6 by at least four log levels was achieved with all disinfection methods (Figure 2B, Appendix A). Interestingly, despite strong reduction in viability of Φ6, no change in the Ct values was observed for the alcoholic surface disinfectant, with mean Ct values of 17.18 and 16.99 before and after the disinfection, respectively. For UV-C only a slight shift was observed for the high dosage with a mean Ct value of 18.25 after radiation with 200 mJ/cm^2^. In contrast, disinfection using ozone resulted in a strong shift with mean Ct values of 29.95 and 28.01 for surfaces exposed directly or difficult to access in a closed container, respectively.

## 4. Discussion

Since the onset of the pandemic, a large number of studies have detected SARS-CoV-2 RNA on inanimate surfaces [7]. In our intensive care unit, we were able to detect viral RNA on surfaces frequently touched by patients and health care workers (HCW) (Figure 1). In patients with high viral load in the upper airways (as detected by quantitative PCR) viral RNA was observed on surfaces touched by HCW, indicating a possible cross-contamination between patients. Therefore, it seems reasonable to reduce viral contamination by surface disinfection in the hospital setting. This assumption was supported by field-setting studies during other coronavirus outbreaks, which showed contaminated surfaces even after terminal cleaning [5]. However, we were unable to cultivate the virus in selected samples with relative high viral load indicating that not every detection of viral RNA must be associated with infectivity. These data fit with results of other studies in which the cultivation of SARS-CoV-2 from inanimate surfaces was successful only in rare cases [7]. Interestingly, the results for SARS-CoV-2 differ markedly from those for viruses with higher tenacity on surfaces. For example, in the recent monkeypox outbreak, most studies attempting to culture the virus from inanimate surfaces in the vicinity of affected patients successfully cultured the virus [12,13,14,15]. These differences indicate that PCR should not be used alone to assess the infectivity of viruses on inanimate surfaces.

During manual terminal disinfection of patient rooms, errors can occur to a relevant extent (up to 50%), as a result of which surfaces are not completely disinfected [16]. To overcome these problems and prevent environment-borne transmission, the usage of automated room disinfection systems for terminal cleaning could be an additional method of disinfection in hospital settings [17,18,19]. In the current pandemic, shortages occurred in the supply of disinfectants [20]. For this reason, we investigated a fully automatic room disinfection system in addition to two well-established disinfectants active against enveloped viruses. The system is able to generate ozone from atmospheric oxygen on site and requires clean water as the only consumable.

The disinfection process using ozone as disinfecting agent resulted in a stronger decrease in SARS-CoV-2 viral RNA compared to UV-C radiation or surface disinfection with a commercial alcoholic disinfectant (Figure 2A; Appendix A). However, the change in Ct values, which can be translated to copies of viral genomes, does not reflect the reduction rate required to claim an agent as disinfectant (4 log reduction to claim virucidal activity). While storage of the surfaces showed no reduction in SARS-CoV-2 viral RNA a detectable reduction in viral RNA was observed for intensive UV-C radiation (log reduction 0.833 to 1.073) and ozone (log reduction 1.073 to 1.835). These data corroborate the observation of Dubuis et al. that ozone combined with high relative humidity is an effective disinfectant for respiratory viruses [21]. Surface disinfection with an alcoholic disinfectant displayed the lowest calculated reduction with a maximum of 0.704 log reduction.

Quantitative results in real-time PCR are highly dependent on the method used for RNA/DNA extraction, the primers used for detection of different genomic targets, the buffers and enzymes used, as well as the analysis method (second derivative or fit points). In a round-robin analysis for quantitative SARS-CoV-2 PCR from defined specimens a large variance of Ct values for identical specimens was observed [22]. Thereby, the amplitude of the variance differed between the targeted genomic regions. Quantification of the viral RNA copies with the use of standards might increase the comparability between laboratories. However, differences caused by the sampling technique will still be present.

In order to correlate changes in PCR Ct values with infective virions we used bacteriophage Φ6, a well-accepted surrogate for enveloped viruses including SARS-CoV-2 [23]. As observed for SARS-CoV-2 from clinical specimens, almost no change in the Ct values was observed during the time of experiments. The stability of this enveloped bacteriophage on inanimate surfaces during this phase was confirmed in the plaque assay with only a slight reduction of 42% (log_10_ = 0.213) over time. Effectivity of disinfection for ozone and an alcoholic surface disinfectant was confirmed with more than 99.9999% reduction in viable bacteriophage (log_10_ = 6.076). For UV-C a dose-dependent reduction of more than 99.999% (log_10_ = 5.045) for a mean intensity (200 mJ/cm^2^) and 99.97% reduction for a low intensity (50 mJ/cm^2^) was found. In contrast to the confirmed inactivation, the Ct values remained unchanged after disinfection with alcohol and only a very slight effect was observed for the disinfection with UV light. Corroborating the observation for surfaces contaminated with SARS-CoV-2 clinical specimen the strongest effects were observed for disinfection with ozone, indicating that ozone effectively degrades the entire virus including the RNA genome.

For PCR detection of Φ6, the amplified region corresponded to a cDNA fragment of 232 bp, so only major degradation of the viral genome would lead to a significant change in Ct values as observed for ozone disinfection. Also, the genomic regions targeted for the detection of human viral pathogens (including SARS-CoV-2) by quantitative PCR are usually less than 300 bp in size. Only major degradation of the genomic RNA will lead to increased Ct values, whereas disruption of the biological membrane or degradation of proteins required for cell entry are sufficient to decrease infectivity. Since the experiments with UV and ozone on artificially contaminated surfaces were performed only on horizontally positioned surfaces, they are not necessarily representative of all surfaces in a room. Especially for UV radiation, a vertical positioning could lead to a higher dose and thus a higher degradation of the RNA.

## 5. Conclusions

The observed lack of significant effects on the detectable viral RNA after effective disinfection without mechanical cleaning suggests that quantitative PCR is not suitable for predicting the infectivity of enveloped viruses (including SARS-CoV-2) on inanimate surfaces. When PCR is performed, only chain breaks in the region of the short amplified genomic region lead to noticeable changes in CT values, so that only more severe degradation of the RNA would result in noticeable changes in this detection method. Therefore, detection of viral genomic fragments on inanimate surfaces by PCR should not be used as a correlate for infectivity.

## Figures and Tables

**Figure 1 ijerph-19-17074-f001:**
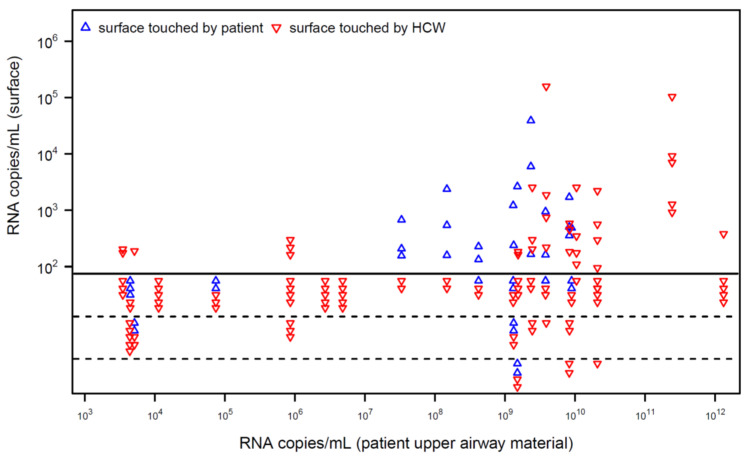
Viral RNA in the immediate patient environment. The number of detectable copies of viral RNA is shown for surfaces in the immediate patient environment. The viral burden on near-patient surfaces in intensive care units was examined on surfaces touched by spontaneously breathing patients (blue triangles) and surfaces used by health care workers (HCW; red triangles) and correlated with the RNA load detectable in the upper airways of the respective patients (nasopharyngeal swab or tracheal secretion) from the same day or with a maximum time interval of one day. Triangles below the solid line represent PCR results below the detection limit. Dashed lines separate the results of different patients with comparable RNA loads in the upper airways.

**Figure 2 ijerph-19-17074-f002:**
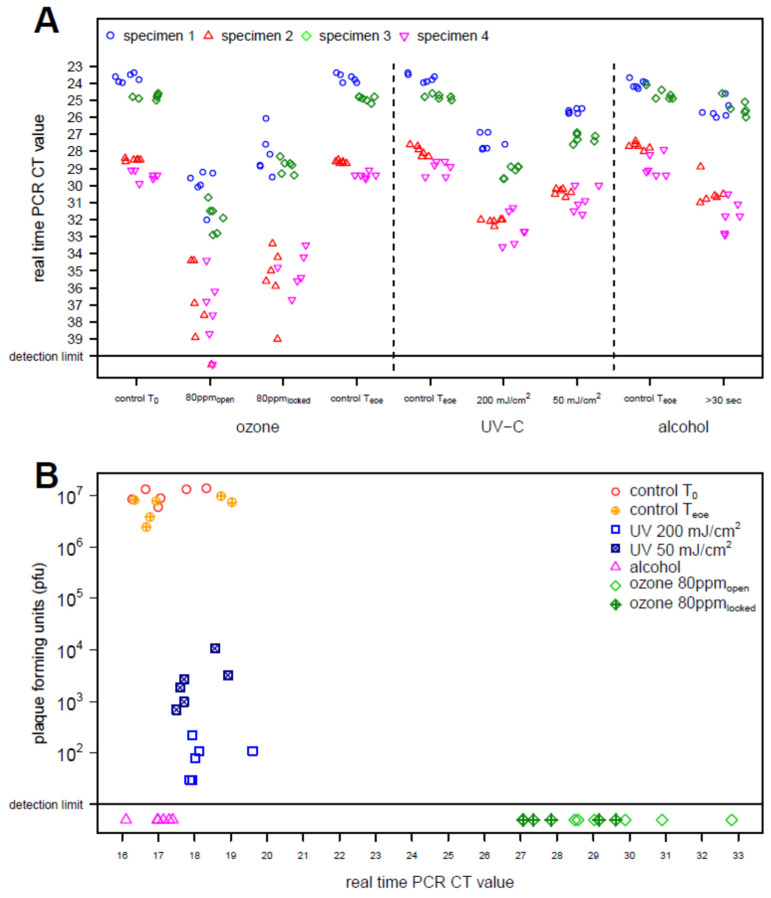
Decrease in Ct values with different disinfection procedures. PCR Ct values for artificially contaminated test surfaces before and after the application of different disinfection methods for SARS-CoV-2 (**A**) and the surrogate enveloped bacteriophage Φ6 (**B**) are displayed. For surfaces artificially contaminated with SARS-CoV-2-positive patient specimen, the shape and color of the marks identify the experiments with materials from four different patients. Marks below the solid line represent PCR results below the detection limit. Dashed lines separate the results of different disinfection methods. Disinfection with ozone was performed openly (80 ppm_open_) in the room or in a gas-tight can with only two 3 mm boreholes (80 ppm_locked_). Due to the long disinfection cycle during ozone disinfection a second control at the end of the experiments was performed (T_eoe_). For disinfection using UV-C, two radiation intensities (200 mJ/m^2^ and 50 mJ/m^2^) were examined. To correlate infectivity with Ct values, untreated surfaces and surfaces treated with the individual methods were compared after contamination with the surrogate enveloped bacteriophage Φ6 (**B**). The surfaces were treated as described for SARS-CoV-2. The shape and color of the marks indicate the different disinfection processes and controls. The Ct values observed with PCR are plotted against the plaque-forming units displaying infectivity of Φ6.

## Data Availability

The data presented in this study are available on request from the corresponding author.

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
