# Peer review of "Infectivity of SARS-CoV-2 on Inanimate Surfaces: Don’t Trust Ct Value"

_ijerph, 2022, doi:10.3390/ijerph192417074_

Round 1

Reviewer 1 Report

Summary -The manuscript by Knobloch and colleagues aimed to increase our understanding of the relationship of recovered CT values to the risk of assessing the potential infectivity hazard of surfaces in the built environment caring for patients infected with the enveloped virus SARS-CoV-2.  Here the authors evaluated surfaces by PCR and culture in 52 patient rooms from an ICU housing COVID-19 patients as well as 53 surfaces artificially contaminated with SARS-CoV-2 for the presence of virus subsequent disinfection by appropriate methods known for their ability to inactivate SARS-CoV-2 and other enveloped viruses.   The results of their study were remarkable in being able to demonstrate that the recovered CT value is not a reflection of whether or not infectious virus was present, as defined by its ability to be cultured, as recovery of culturable virus from the patient care setting was not observed.    The results of manuscript while significant were not remarkable in retrospect as the disinfecting protocols used are not known for their ability to degrade nucleic acids to a sufficient level where a PCR to a limited target would likely fail.  Consequently the results offer to the infection control community that a positive detection SARS-CoV-2 nucleic acid from the built environment should not be considered as a prima facie evidence that the surfaces within the built environment pose an infectious hazard as a fomite for SARS-CoV-2 transmission.  So this reviewer is agreement with the authors that one should not trust Ct values in this context and it is wise to mechanically clean surfaces to ensure sanitation of surfaces, even with soap and warm water, within the built environment to remove enveloped viruses.

Critique I would have like to see if dilute sodium hypochlorite would have been effective at reducing the concentration of recoverable nucleic acid from the surfaces in patient care settings.  However, I commend the authors for attempting to address with Ozone and UV.   It is likely that the target size the effective energy disperal of both the ozone and UV limited the fracturing of the nucleic acid which the authors may wish to comment on in the discusssioin.   

Author Response

Many thanks for the constructive review. Unfortunately, we were not able to perform further experiments in the short time available to revise the manuscript, so we were not able to collect data on dilute sodium hypochlorite. In our study we had only tested procedures which are available as a standard in our hospital.
We adopted the proposal and discussed the limitations to degaradation of RNA by ozone and especially UV in the chosen experimental setup. 
In terms of language, the manuscript was revised according to the suggestions of the second reviewer and after correction by a native speaker.

Reviewer 2 Report

This manuscript describes experiments to recover SARS-CoV-2 from surfaces in the patient environment, and subsequent experiments to demonstrate the lack of relationship between Ct and infectious virus.

In general, I find the work to be scientifically sound, but there are several issues of English usage and data presentation that complicate the understanding of the results. I have outlined them in general in the manuscript, and recommend a more extensive English language review prior to acceptance. I was also unable to access supplementary material, which I believe is critical for the reader. Lastly, the finding that Ct value is not a reliable measure of viable virus isn't a completely novel finding, however, there is still value in the study, in general, due to the novelty and interest in the disinfection methods and experiment itself. In my opinion, the paper should be presented in a way that focuses more on how these disinfection processes impact viability and detection by PCR differently, and how that translates to interpretation of data from the hospital.

Author Response

Many thanks for the constructive review. We have adopted your suggestions and comments and answered them point by point in the attached pdf. Furthermore, we have had an additional correction of the language made by a native speaker. 

Round 2

Reviewer 2 Report

Thank you for consideration of my comments. I find this paper suitable for publication.